# Bayes-MIL: A New Probabilistic Perspective on Attention-based Multiple Instance Learning for Whole Slide Images

Yufei Cui[1], Ziquan Liu[2], Xiangyu Liu[3],
Xue Liu[1], Cong Wang[2], Tei-Wei Kuo[45], Chun Jason Xue[2], Antoni B. Chan[2]
[1]McGill University    [2]City University of Hong Kong    [3]Bingli AI Research
[4]National Taiwan University    [5]Mohamed bin Zayed University of Artificial Intelligence

## Abstract

Multiple instance learning (MIL) is a popular weakly-supervised learning model on the whole slide image (WSI) for AI-assisted pathology diagnosis. The recent advance in attention-based MIL allows the model to find its region-of-interest (ROI) for interpretation by learning the attention weights for image patches of WSI slides. However, we empirically find that the interpretability of some related methods is either untrustworthy as the principle of MIL is violated or unsatisfactory as the high-attention regions are not consistent with experts' annotations. In this paper, we propose Bayes-MIL to address the problem from a probabilistic perspective. The induced patch-level uncertainty is proposed as a new measure of MIL interpretability, which outperforms previous methods in matching doctors annotations. We design a slide-dependent patch regularizer (SDPR) for the attention, imposing constraints derived from the MIL assumption, on the attention distribution. SDPR explicitly constrains the model to generate correct attention values. The spatial information is further encoded by an approximate convolutional conditional random field (CRF), for better interpretability. Experimental results show Bayes-MIL outperforms the related methods in patch-level and slide-level metrics and provides much better interpretable ROI on several large-scale WSI datasets.

## 1 Introduction

In real-world applications of deep learning, data like images or texts are often associated with insufficient labels, due to the expensive annotation cost. For example, the whole slide images (WSI) for medical diagnosis have about $10^5 \times 10^5$ pixels per image, but are tagged with single categorical labels (Zhang et al., 2019; Campanella et al., 2019). Weakly-supervised learning methods are designed for learning representations and making decision in these cases. Multiple instance learning (MIL) is a popular weakly-supervised learning model for the application of WSI recognition (Ilse et al., 2018; Lu et al., 2021). Concretely, a large WSI slide is sliced into a bag of image patches (instances) with a moderate size.[1] MIL builds an end-to-end parametric model that aggregates the learned features from instances and only learns from bag-level labels. The rule of aggregation is implementing *the key principle of MIL*: for binary classification, a bag is negative when all instances are negative, and a bag is positive when there is one or more positive instance (Ilse et al., 2018).

Recent advances study the *attention-based MIL* for re-weighing the instances for better performance. This attention mechanism for MIL is extensively explored and used as a measure of interpretability in various downstream tasks for medical diagnosis, like prostatic cancer (Zhang et al., 2021), breast cancer (Naik et al., 2020), etc. Specifically, the high attention weights are used to indicate that its associated instances are positive instances, e.g, the cancerous image patches. However, this rule is not formally justified and it is not clear whether the negative instances (i.e., benign) would be assigned a high attention value or the other way around. We first analyze the convergence of attention and provide validity of this rule under binary labels. Based on this rule, we conduct an empirical study on a large scale WSI dataset for how the attention mechanism in the related MIL methods performs. The study shows two clear flaws of the related methods:

---

[1]We define the following interchangeable terms for simplicity: "bag" and "slide"; "instance" and "patch".

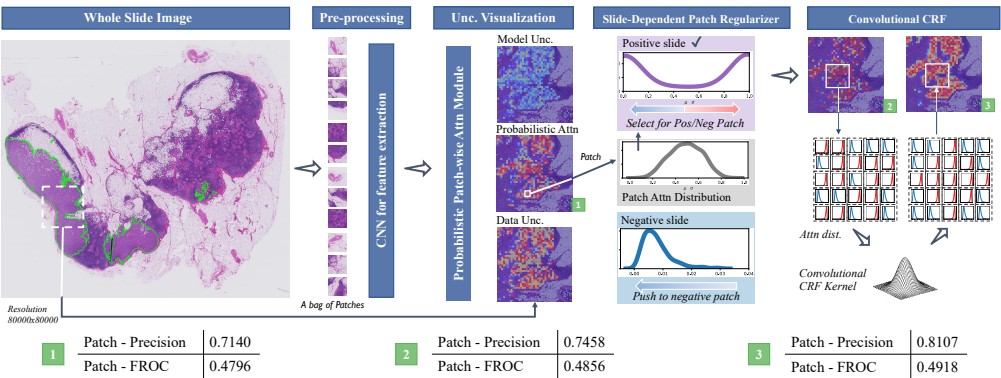

Figure 1: The overview of our Bayes-MIL framework and zoom-in views for clear visualization of interpretability. **(1)** The basic Bayes-MIL improves patch-level localization performance (Sec. 3.1). **(2)** The slide-dependent patch regularizer makes attention densely concentrated on the positive area, improving its interpretability (Sec. 3.2). **(3)** The convolutional CRF improves the localization by smoothing the uncertainty over different patches (Sec. 3.3). (bottom) The ablation results on a few metrics show the improvement of interpretability. The full ablation results are in Sec. 5.

- The interpretability for negative bags is untrustworthy because some methods violate *the key principle of MIL* by placing high attention values on negative bags, thus indicating positive instances.
- The interpretability for positive bags is unsatisfactory because the high attention values could not well match experts' annotations of positive instances.

In this paper, we address the problems from a probabilistic perspective. First, a basic framework of Bayesian MIL (Bayes-MIL) is proposed, for inducing uncertainty over the attention weights. The uncertainty is potentially an accurate measure for guessing whether the instances are positive or negative, as a replacement of attention. Second, a regularizer is designed by deduction from the MIL principle and implemented via the variational inference framework, which sets specific constraints for the attention distributions of positive and negative bags. Third, to encode the spatial information of instances for medical imaging application, we propose an approximate operation to the convolutional conditional random field, which benefits the localization of the region of interest (ROI). The final classifier is modeled in a Bayesian way, in order to provide calibrated uncertainty of the bag-level prediction. The overview of our proposed method is shown in Fig. 1. The contributions of this paper are listed as follows:

- We analyze the attention-based MIL on the interpretability-critic medical application and point out the flaws by directly using attention for interpretation.
- To address these problems, we propose the first Bayesian MIL for WSI with 3 key components: a probabilistic instance-wise attention module for uncertainty visualization, the slide-dependent patch regularizer for learning the correct attention distribution, and an approximate convolutional conditional random field for encoding spatial information. Our model provides well-calibrated uncertainties, which is crucial for safety in medical applications.
- The evaluation on large-scale MIL datasets shows Bayes-MIL outperforms the related methods in instance-level interpretation and bag-level prediction under various evaluation metrics. The visualized distribution of data uncertainty shows a strong correlation of the designed regularizer, which validates the soundness of regularizer and explains why uncertainty is useful in MIL interpretation.

## 2 FORMULATION AND ANALYSIS OF MULTIPLE INSTANCE LEARNING

**Multiple instance learning formulation** We follow the standard formulation of Attention-based Multiple Instance Learning (MIL) (Ilse et al., 2018; Lu et al., 2021). In MIL, the input is a bag of instances, $\boldsymbol{X} = \{\boldsymbol{x}_1, \ldots, \boldsymbol{x}_K\}, \boldsymbol{x}_k \in \mathbb{R}^D$. $K$ is the number of instances, which varies for different bags. There is a bag-level label $Y$. We further assume the instances also have corresponding instance-level labels $\{y_1, \ldots, y_K\}$, which are *unknown* during training. There are $N$ such bag-label pairs constituting the dataset $\mathcal{D} = \{\boldsymbol{X}_n, Y_n\}_{n=1}^N$. The objective of MIL is to learn an optimal function for predicting the bag-level label with the bag of instances as input. To this end, the MIL model should be able to aggregate the information of instances $\{\boldsymbol{x}_k\}_{k=1}^K$ to make the final decision. A well-adopted aggregation method is the embedding-based approach which maps $\boldsymbol{X}$ to a bag-level representation $\boldsymbol{z} \in \mathbb{R}^D$ and use $\boldsymbol{z}$ to predict $Y$. Ilse et al. (2018) extends the embedding-based

aggregation approach by leveraging the attention mechanism, namely attention-based deep MIL (ABMIL). First, a transformation $g(\cdot)$ computes a low-dimensional embedding $\boldsymbol{h}_k = g(\boldsymbol{x}_k) \in \mathbb{R}^D$ for each instance $\boldsymbol{x}_k$. The attention module aggregates the set of embeddings $\{\boldsymbol{h}_k\}_{k=1}^K$ into a bag level embedding $\boldsymbol{z}$, $\boldsymbol{z} = \sum_{k=1}^K a_k \boldsymbol{h}_k$, where the attention for the $k$-th instance is computed via a softmax function,

$$a_k = f_{\boldsymbol{\pi}}(\boldsymbol{H})_k = \frac{\exp\{\boldsymbol{m}^T(\tanh(\boldsymbol{V}_1^T \boldsymbol{h}_k) \odot \text{sigmoid}(\boldsymbol{V}_2^T \boldsymbol{h}_k))\}}{\sum_{j=1}^K \exp\{\boldsymbol{m}^T(\tanh(\boldsymbol{V}_1^T \boldsymbol{h}_j) \odot \text{sigmoid}(\boldsymbol{V}_2^T \boldsymbol{h}_j))\}}. \tag{1}$$

where the attention function $f_{\boldsymbol{\pi}} : \mathbb{R}^{D \times K} \to \mathbb{R}^K$ with parameters $\boldsymbol{\pi} = \{\boldsymbol{m}, \boldsymbol{V}_1, \boldsymbol{V}_2\}$, and $f_{\boldsymbol{\pi}}(\boldsymbol{H})_k$ denotes the $k$-th output of $f$. $\boldsymbol{H} = [\boldsymbol{h}_1, \ldots, \boldsymbol{h}_K] \in \mathbb{R}^{D \times K}$ is the matrix of embeddings. The bag embedding $\boldsymbol{z}$ is then mapped to the logits $\boldsymbol{u}$ with a feed forward layer with parameter $\boldsymbol{W}$ for the bag-level classification, $\boldsymbol{u} = \boldsymbol{W}^T \boldsymbol{z}$.

**Multiple instance learning for medical imaging**    WSI is a type of high-dimensional image data format (up to $10^5 \times 10^5$ pixels per image) widely adopted in the medical area. Due to the scarcity of experts and the high annotation cost, only class labels (e.g., diagnosis results) are available for most WSIs. The high resolution of data and lack of precise annotations raise challenges for machine learning-assisted classification. To fit the data in modern computation hardware, the high-resolution image slides (*slide*) are partitioned into image patches (*patch*) before further processing. MIL fits the classification task of WSI by corresponding the slides to bags, and patches to instances. The transformation $g(\cdot)$ is the feature extractor from a pre-trained convolutional neural network. The MIL model only predicts the slide-level label, e.g., whether a slide is cancerous or not. However, the interpretation of the patches is crucial, e.g., which patches indicate the cancer, as users always check the interpretation before trusting the prediction. ABMIL uses the attention weights to tell which patches the MIL model focuses on, and several works DSMIL (Li et al., 2021), CLAM (Lu et al., 2021), TransMIL (Shao et al., 2021) study the variants of attention for better interpretability.

Ilse et al. (2018) suggested that, with binary classification label $Y \in \{0, 1\}$, the high attention weights in ABMIL could locate the positive area ($y_k = 1$) in an ideal case. The followup works from ABMIL use the high attention weights to indicate the positive patches in a black-box manner. However, there is no formal justification for this claim, and it is still unclear whether a high attention weight could also be assigned to a negative area during training. Therefore, we analyze the general attention-based MIL framework to provide this justification.

**The convergence of attention**    Assume that the $j$th input patches in $i$th slide is $\boldsymbol{h}_{ij}$, the classifier weight is $\boldsymbol{w} \in \mathbb{R}^D$, the attention variable is $\boldsymbol{a} \in (0, 1)^{K \times 1}$. The output of the network $\hat{y}_i$ and loss function $\mathcal{L}_i$ for the $i$-th slide are

$$\hat{y}_i = \sigma(\boldsymbol{w}^T \boldsymbol{H}_i \boldsymbol{a} + b), \quad \mathcal{L}_i = -Y_i \log(\hat{Y}_i) - (1 - Y_i) \log(1 - \hat{Y}_i). \tag{2}$$

The $\boldsymbol{H}_i$ is all patches in $i$th slide, where the $l_2$ norm of all $h_{i,j}$ is upper bounded by 1, and $\sigma(x) = 1/(1 + \exp(-x))$ is the sigmoid function. Assume the positive and negative patches are linear separable and there is an optimal $\boldsymbol{w}^*$, where $\|\boldsymbol{w}^*\| = 1$ and the margin is $\gamma = \min_{i,j} |\boldsymbol{w}^{*T} \boldsymbol{h}_{i,j}|$. If a slide $\boldsymbol{H}_i$ is negative, then all patches in $\boldsymbol{H}_i$ are negative, i.e., $\boldsymbol{w}^{*T} \boldsymbol{h}_{i,j} < 0, \forall j$ if $Y_i = 0$. If a slide $\boldsymbol{H}_i$ is positive, then the first $K_p$ patches are positive and the last $K_n$ patches are negative, i.e., $\boldsymbol{w}^{*T}[\boldsymbol{h}_{i,1}, \cdots, \boldsymbol{h}_{i,K_p}] > \boldsymbol{0}$ and $\boldsymbol{w}^{*T}[\boldsymbol{h}_{i,K_p+1}, \cdots, \boldsymbol{h}_{i,K_p+K_n}] < \boldsymbol{0}$ if $Y_i = 1$. There is an optimal $\boldsymbol{a}^*$ so that the first $K_p$ dimension is $(1 - \epsilon)/K_p$ and the last dimension is $\epsilon$, where $\epsilon$ is an infinitesimal, e.g., 1e-5. For the initialization of $\boldsymbol{a}$, we assume $\boldsymbol{w}^{*T} \boldsymbol{H}_i \boldsymbol{a} > \zeta > 0, \forall i$. The $\boldsymbol{a}$ is output of softmax function, $\boldsymbol{a} = s(\boldsymbol{u})$, where $a_j = \frac{\exp(u_j)}{\sum_{k=1}^K \exp(u_k)}$.

**Lemma 1.** *If we train the $\boldsymbol{w}$ and $\boldsymbol{a}$ as described in Appendix 1, the $\boldsymbol{w}$ converges to the optimal $\boldsymbol{w}^*$ in at most $4/max(\gamma, \zeta)^2$ steps and $\boldsymbol{a}$ converges to the desired $\boldsymbol{a}^*$, where the first $K_p$ elements are large and the last $K_n$ elements are small.*

Lemma 1 indicates that MIL is guaranteed to converge to the desired attention variable under the ideal condition. Note that in reality, the position of positive patches is not fixed so we use a parameterized function to make the attention variable $a_j$ depend on the patch feature $\boldsymbol{h}_{i,j}$. Lemma 1 provides a guarantee for the validity of visualization and the design of our framework. However, in reality, the convergence of weight and attention variables depends on their initialization. Thus, existing methods may not have a good match between high-attention patches and ground-truth positive patches. The following empirical study is performed to illustrate this.

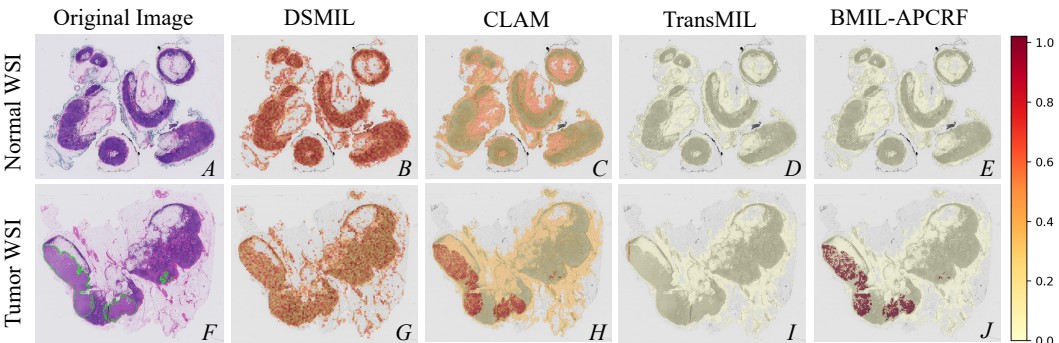

Figure 2: The visualization of normal and tumor slides and the ROIs provided by different models. The patch-level annotations for the tumor image are shown in green color in (F). The attention values $\boldsymbol{a}$ are normalized to the same range by $\frac{\boldsymbol{a}-\min_{\boldsymbol{a}}}{\max_{\boldsymbol{a}}-\min_{\boldsymbol{a}}}$. The $\min_{\boldsymbol{a}}$ and $\max_{\boldsymbol{a}}$ are the same for all methods for better visualization.

**Empirical study of interpretability**  We use the attention $\{a_k\}_{k=1}^K$ as the patch-level prediction confidence and compare with the patch-level ground-truth $\{y_k\}_{k=1}^K$, using the related MIL approaches. One key discovery from the related methods is that *high attention values are still generated, indicating positive patches, for negative slides*, as shown in Fig. 2B 2C 2D. Another discovery is that the high attention values are not concentrated on the doctor's annotation well during inference time, as shown in Fig. 2G 2H 2I (see numerical results in Tab. 1).

**The need for Bayesian modelling of MIL**  The empirical study constitutes one motivation for our probabilistic approach: bring more stochasticity to the optimization process so that the convergence of weight and attention variables does not heavily depend on the initialization. Thus, we study the probabilistic counterpart of MIL, namely Bayes-MIL, which is further potentially beneficial in the following three aspects:

- *Better optimization:* Besides stochasticity in optimization, by turning $\boldsymbol{a}$ to stochastic nodes, explicit regularization could be imposed for generating correct attention (see Fig. 2E 2J).
- *New measure of interpretability:* By learning a proper posterior over the parameters $p(\boldsymbol{\pi}|\mathcal{D})$, we can induce the uncertainty over patches by $p(\boldsymbol{a}|\boldsymbol{H}^*,\mathcal{D}) = \int p(\boldsymbol{a}|\boldsymbol{\pi},\boldsymbol{H}^*)p(\boldsymbol{\pi}|\mathcal{D})d\boldsymbol{\pi}$, where $\boldsymbol{H}^*$ is the testing data. Patch-level uncertainty can potentially indicate which patches the model is uncertain about, becoming a new measure of interpretability for MIL. In other words, the patch-level uncertainty is leveraged for localizing positive areas.
- *Calibrated uncertainty:* A properly learned posterior $p(Y|\boldsymbol{H}^*,\mathcal{D})$ provides a *calibrated uncertainty* on $Y$, which is crucial in the application of medical imaging but ignored in existing methods.

# 3  BAYESIAN MULTIPLE INSTANCE LEARNING

## 3.1  INSTANCE-LEVEL DISENTANGLED UNCERTAINTY

The basic framework of Bayes-MIL is introduced in this section. To obtain uncertainty over the attention, the principled way is to assume a prior distribution $p(\boldsymbol{\pi})$ on the parameters of attention function $f_{\boldsymbol{\pi}}(\cdot)$, and let the model learn a posterior $p(\boldsymbol{\pi}|\mathcal{D})$. The other way is to directly learn an empirical posterior distribution $p(\boldsymbol{\pi}|\mathcal{D})$, e.g., ensembles. The posterior induces the uncertainty over the attention by $p(\boldsymbol{a}|\boldsymbol{H}^*,\mathcal{D}) = \int p(\boldsymbol{a}|\boldsymbol{\pi},\boldsymbol{H}^*)p(\boldsymbol{\pi}|\mathcal{D})d\boldsymbol{\pi} \approx \frac{1}{S}\sum_{s=1}^S f_{\boldsymbol{\pi}_s}(\boldsymbol{H}^*), \boldsymbol{\pi}_s \sim p(\boldsymbol{\pi}|\mathcal{D})$, where the attention function directly models the conditional distribution, i.e., $p(\boldsymbol{a}|\boldsymbol{\pi},\boldsymbol{H}) = f_{\boldsymbol{\pi}}(\boldsymbol{H}) = f(\boldsymbol{\pi},\boldsymbol{H})$.

For making $p(\boldsymbol{a}|\boldsymbol{\pi},\boldsymbol{H})$ a strict distribution, the softmax function in (1) could be leveraged for normalization, $\sum_k a_k = 1$. Then, $\boldsymbol{a}$ is a vector over the simplex, representing one categorical distribution. However, in this case, we could only calculate the uncertainty for the single categorical distribution. To extract the patch-level uncertainty, there should be one probabilistic distribution for the attention of each patch. Therefore, we need to model the distribution for patches, $p(a_k|\boldsymbol{\pi},\boldsymbol{H}) = f_{\boldsymbol{\pi}}(\boldsymbol{H})_k = f(\boldsymbol{\pi},\boldsymbol{H})_k$ and normalize for each patch. To this end, we replace the softmax function in (1) by an element-wise sigmoid function,

$$a_k = \frac{1}{1+\exp\{-\boldsymbol{m}^T(\tanh(\boldsymbol{V_1}\boldsymbol{h}_k^T)\odot\mathrm{sigmoid}(\boldsymbol{V_2}\boldsymbol{h}_k^T))\}}, \quad \boldsymbol{z} = \frac{1}{\sum_k a_k}\sum_k a_k\boldsymbol{h}_k \qquad (3)$$

where the parameters $\boldsymbol{\pi} = \{\boldsymbol{m},\boldsymbol{V_1},\boldsymbol{V_2}\}$ are shared across all patches. The second normalization when computing $\boldsymbol{z}$ is for numerical stability.

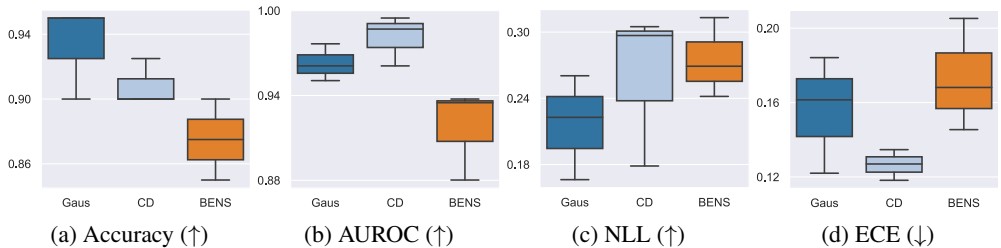

Figure 3: The comparisons of multiplicative Gaussian, concrete dropout and BatchEnsemble for modelling probabilistic MIL weights. The slide-level results are reported. NLL is negative log-likelihood and ECE is the expected calibration error for measuring uncertainty calibration.

With this treatment, the patch-level model uncertainty and data uncertainty for each patch $a_k$ could be extracted by (Houlsby et al., 2011; Gal et al., 2017b):

$$\underbrace{\mathcal{I}[a_k, \boldsymbol{\pi}|\boldsymbol{H}^*, \mathcal{D}]}_{\text{Model Uncertainty}} = \underbrace{\mathcal{H}[\mathbb{E}_{p(\boldsymbol{\pi}|\mathcal{D})}[p(a_k|\boldsymbol{\pi}, \boldsymbol{H}^*)]]}_{\text{Total Uncertainty}} - \underbrace{\mathbb{E}_{p(\boldsymbol{\pi}|\mathcal{D})}[\mathcal{H}[p(a_k|\boldsymbol{\pi}, \boldsymbol{H}^*)]]}_{\text{Data Uncertainty}} \quad (4)$$

We explore three different methods for modeling the probabilistic weights $\boldsymbol{\pi}$: Batch-Ensemble (Wen et al., 2020; Dusenberry et al., 2020) as the empirical posterior, Concrete Dropout (Gal et al., 2017a) and multiplicative Gaussian noise (Kingma et al., 2015; Molchanov et al., 2017; Cui et al., 2021) that assume prior on $\boldsymbol{\pi}$ and derive the posterior based on variational inference. As there is no direct indication of which set of posterior and prior, we should choose, we validate them empirically with the proposed framework. The experiments are conducted on CAMELYON16 dataset with 5 splits of training and validation sets. For the Batch-Ensemble, we use an ensemble of size 4. For Concrete Dropout and multiplicative Gaussian noise, we take 4 samples from the learned posterior during inference time. The multiplicative Gaussian method is selected for the probabilistic weights due to a high accuracy and a low NLL, shown in Fig. 3.

For neural networks, computing the posterior distribution using the Bayes rule requires computing intractable integrals over $\boldsymbol{\pi}$. In this paper, to be consistent with the stochastic attention function modelling, we use variational inference for approximating the posterior. Specifically, the posterior $p(\boldsymbol{\pi}|\mathcal{D})$ is approximated by $q_{\phi}(\boldsymbol{\pi})$, by minimizing the Kullback-Leibler (KL) divergence $\text{KL}[q_{\phi}(\boldsymbol{\pi})||p(\boldsymbol{\pi}|\mathcal{D})]$, where $\phi$ are the variational parameters. This is equivalent to maximizing the *evidence lower bound*:

$$\max_{\phi} L_{\phi} = L_{\mathcal{D}}(\boldsymbol{\phi}) - \text{KL}[q_{\phi}(\boldsymbol{\pi})||p(\boldsymbol{\pi})], \quad L_{\mathcal{D}}(\boldsymbol{\phi}) = \sum_{i=1}^{N} \mathbb{E}_{q_{\phi}(\boldsymbol{\pi})}[\log p(Y_i|\boldsymbol{H}_i, \boldsymbol{\pi})], \quad (5)$$

where $L_{\mathcal{D}}(\boldsymbol{\phi})$ is the expected data log-likelihood and $p(\boldsymbol{\pi})$ is the prior over $p(\boldsymbol{\pi})$.

## 3.2 SLIDE-DEPENDENT PATCH REGULARIZER

Although the proposed model can naturally visualize different types of uncertainty at the patch level and better localize the ROI, we can further *leverage the slide-level information for building a strong regularization on the ROI localization*. The aim is to explicitly encode the underlying logic of MIL into the training process, instead of letting the model explore implicit decision rules during training.

Recall $Y$ is the slide-level label and $\{y_k\}_{k=1}^{K}$ are the labels for patches which are unknown during training. The intuition is based on the logic under the MIL framework that a slide is negative when all patches are negative, while a slide is positive when there are one or more positive patches. For binary label, $Y = 0$ iff $\sum_k y_k = 0$, and $Y = 1$ otherwise. The following design principle can be drawn by simple deduction: When $Y = 0$, the attention distributions $\{p(a_k|\boldsymbol{\pi}, \boldsymbol{H})\}_{k=1}^{K}$ must concentrate on the negative side ($a_k = 0$) to guarantee $y_k = 0$. When $Y = 1$, the attention distributions are free to select either the positive ($a_k = 1$) or negative sides. However, for precise localization, the attention distributions must concentrate on either the positive or negative side with high confidence.

This design principle is implemented by a variational inference framework by finding a regularizer on patches that is dependent on the slide label. Specifically, we choose the logit-normal distribution for $a_k$, due to its expressiveness over the simplex. The regularizer (a non-strict prior) is defined as

$$p(a_k|Y) = (1-Y)\mathcal{LN}(\mu_0, \sigma_0) + Y\mathcal{LN}(\mu_1, \sigma_1), \quad (6)$$

where $\mathcal{LN}(a_k|\mu, \sigma) = \frac{1}{\sigma\sqrt{2\pi}} \frac{1}{a_k(1-a_k)} e^{-\frac{\text{logit}(a_k)-\mu}{2\sigma^2}}$ is the logit-normal distribution. $\{\mu_0, \sigma_0\}$ and $\{\mu_1, \sigma_1\}$ are the pairs of mean and variance for the negative slides and the positive slides, respectively. As shown in Fig. 4, by setting the parameters for the regularizer, we implement the design

principle: $p(a_k|\mu_0, \sigma_0)$ concentrates on the negative side and $p(a_k|\mu_1, \sigma_1)$ allows selecting either the negative or the positive side with high confidence.

For the MIL model, we divert $\{a_k\}_{k=1}^K$ from induced distributions to stochastic nodes. The approximate posterior is defined as $q(a_k|\boldsymbol{\mu}, \boldsymbol{\sigma}) = \mathcal{LN}(\mu_k = f_\mu(\boldsymbol{\pi}, \boldsymbol{H})_k, \sigma_k = f_\sigma(\boldsymbol{\pi}, \boldsymbol{H})_k)$ and $\sum_k \mathrm{KL}[q(a_k|\boldsymbol{\mu}, \boldsymbol{\sigma})||p(a_k|Y)]$ is used as a regularization term during training, which we denote as the slide-dependent patch regularizer (SDPR). When a negative slide is given, the mode of the attention posterior is pushed to the negative side, generating low attention values over all patches. When a positive slide is given, the posterior will be trained to select a positive mode or a negative mode, with only high concentration. This produces dense localization of the ROI. The derivation of KL divergence for the regularization is in the Appendix.

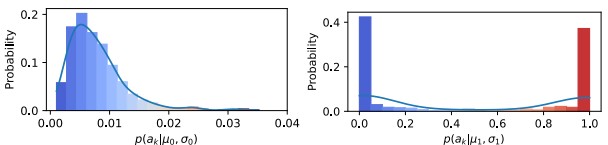

Figure 4: The visualization of density (curves) of the regularizer for the (left) negative and (right) positive slides. Samples bars are visualized with color based on the attention value.

The SDPR is also beneficial for improving the slide-level performance of MIL. The reason is it explicitly constrains the model to generate low attention values for all patches from the negative slides. Note that the major obscurity in MIL comes from the positive slides, where positive patches and negative patches coexist.

There is *no obscurity* for negative slides as all patches are negative by definition. However, the previous MIL models are still free to generate high attention for patches from negative slides, which neglect the underlying logic of MIL.

## 3.3 ENCODING SPATIAL INFORMATION VIA APPROXIMATION TO CONVOLUTIONAL CRF

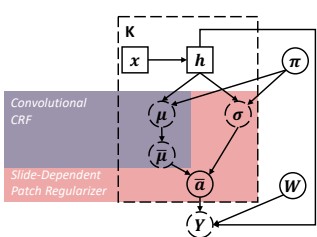

Figure 5: The graphical representation of Bayes-MIL. Boxes are deterministic nodes. Circles are stochastic nodes. Dashed circles are nodes of induced distributions.

The spatial information between patches are important for modelling of MIL for WSI recognition and localization, as there exist patches are spatially correlated (Shao et al., 2021). Here, we study how to encode the spatial information from a Bayesian perspective. The intuition is to let the neighboring attention posteriors $q(a_k)$ influence each other.

One principled method for encoding the spatial information is to use conditional random field (CRF). Assume $\bar{\boldsymbol{a}}$ is the output attention variable of CRF and $\boldsymbol{\Theta}$ is the input variable in CRF, from data. With CRF, the distribution of $p(\bar{\boldsymbol{a}}|\boldsymbol{m})$ could be directly modelled with $\boldsymbol{m}$ as the spatial features of patches (Zheng et al., 2015; Teichmann & Cipolla, 2018).

$$p(\bar{\boldsymbol{a}}|\boldsymbol{m}) = \frac{1}{Z(\boldsymbol{m})}e^{-E(\bar{\boldsymbol{a}}|\boldsymbol{m})}, \tag{7}$$

$$E(\bar{\boldsymbol{a}}|\boldsymbol{m}) = \sum_k \psi_u(\bar{a}_k|\boldsymbol{m}) + \sum_{k\neq j} \psi_p(\bar{a}_k, \bar{a}_j|\boldsymbol{m})$$

where $\boldsymbol{m} = [\boldsymbol{\omega}, \boldsymbol{\eta}, \boldsymbol{a}]$ contains the coordinates of the each patch over the slide, $\boldsymbol{\omega} = [\omega_1, \ldots, \omega_K]^T$ and $\boldsymbol{\eta} = [\eta_1, \ldots, \eta_K]^T$. $\psi_u$ is the unary potential that contains information only from the single patch and $\psi_p$ is the pair-wise potential that captures the pair-wise correlation between patches. However, calculating the pair-wise correlation between patches has $\mathcal{O}(K^2)$ complexity. For efficiency, we only consider the local dependency around $a_k$, by setting the non-local pair-wise correlation to be 0. This is equivalent to applying a softmax normalization and a convolution operation for the provided input $\boldsymbol{a}$. We define an function for the proposed convolutional CRF, $\bar{\boldsymbol{a}} = C_{\boldsymbol{w},\boldsymbol{h}}(\boldsymbol{a})$. Specifically, $\tilde{\boldsymbol{a}} = \mathrm{softmax}(\boldsymbol{a})$, $\hat{\boldsymbol{a}} = \mathrm{reshape}(\boldsymbol{w}, \boldsymbol{h}, \tilde{\boldsymbol{a}})$, $\bar{\boldsymbol{a}} = \mathrm{convolution}(\hat{\boldsymbol{a}}, \mathcal{K})$, where $\mathcal{K}$ is a convolutional kernel with predefined hyperparameters. The detailed derivations and algorithms for the full convolutional CRF for BayesMIL are in the Appendix. For a precise estimation of the variable $\bar{\boldsymbol{a}}$, the following Monte-Carlo estimator is required,

$$\mathbb{E}[\bar{\boldsymbol{a}}] = \mathbb{E}_{q(\boldsymbol{a}|\boldsymbol{\mu},\boldsymbol{\sigma})}[C_{\boldsymbol{w},\boldsymbol{h}}(\boldsymbol{a})] \approx \frac{1}{S}\sum_{s=1}^S [C_{\boldsymbol{w},\boldsymbol{h}}(\boldsymbol{a}_s)], \; \boldsymbol{a}_s \sim q(\boldsymbol{a}|f_\mu(\boldsymbol{\pi}, \boldsymbol{H}), f_\sigma(\boldsymbol{\pi}, \boldsymbol{H})) \tag{8}$$

The full convolutional CRF has the most promising results on real-world datasets. However, the repetitive sampling makes the training less efficient. To bypass the heavy-load of sampling, we use

a first-order Taylor expansion to approximate the expectation (see Appendix). The final format of approximation could be written as

$$q(\bar{a}_k|\bar{\boldsymbol{\mu}} = C_{\boldsymbol{w},\boldsymbol{h}}(\boldsymbol{\mu}), \boldsymbol{\sigma}) = \mathcal{LN}(\bar{\mu}_k = C_{\boldsymbol{w},\boldsymbol{h}}(f_\mu(\boldsymbol{\pi},\boldsymbol{H}))_k, \sigma_k = f_\sigma(\boldsymbol{\pi},\boldsymbol{H})_k), \tag{9}$$

for imposing slide-level patch regularizer, as well as uncertainty disentanglement and visualization.

### 3.4 PUTTING IT ALL TOGETHER

The graphical model for the Bayes-MIL is shown in Fig. 5. For the classifier with parameter $\boldsymbol{W}$, we also use a multiplicative Gaussian modelling (Kingma et al., 2015) as the posterior and posterior for calibrated uncertainty. The final objective is:

$$\max_{\phi} \quad L_{\mathcal{D}}(\phi) - \lambda_0 R_{\boldsymbol{\pi},\boldsymbol{W}}(\phi) - \lambda_1 R_{\bar{\boldsymbol{a}}}(\phi) \tag{10}$$

$$L_{\mathcal{D}}(\phi) = \sum_{i=1}^{N} \mathbb{E}_{q_\phi(\boldsymbol{\pi},\boldsymbol{W})}[\log p(Y_i|\boldsymbol{H}_i, \boldsymbol{\pi}, \boldsymbol{W})], \quad R_{\boldsymbol{\pi},\boldsymbol{W}}(\phi) = \mathrm{KL}[q_\phi(\boldsymbol{\pi},\boldsymbol{W})||p(\boldsymbol{\pi},\boldsymbol{W})],$$

$$R_{\bar{\boldsymbol{a}}}(\phi) = \sum_{i=1}^{N}\sum_{k=1}^{K} \mathrm{KL}[q(\bar{a}_k|C_{\boldsymbol{w},\boldsymbol{h}}(f_\mu(\boldsymbol{\pi},\boldsymbol{H}_i)), f_\sigma(\boldsymbol{\pi},\boldsymbol{H}_i))||p(\bar{a}_k|Y_n)]$$

where $R_{\boldsymbol{\pi},\boldsymbol{W}}(\phi)$ is KL term for the probabilistic weights and $R_{\bar{\boldsymbol{a}}}(\phi)$ is the SDPR for the attention. $\lambda_0$ and $\lambda_1$ are the trade hyperparameters for two terms respectively.

## 4 RELATED WORK

**Attention-based multiple instance learning** For improving interpretability and performance of multiple instance learning (MIL), ABMIL (Ilse et al., 2018) first introduces the attention mechanism for embedding-based MIL. DSMIL (Li et al., 2021) considers contrastive learning for feature extraction and builds the global connection between patch attentions. TransMIL (Shao et al., 2021) proposes a correlated MIL and implements it by multi-head self-attention and spatial information encoding for full global correlation. CLAM (Lu et al., 2021) extends ABMIL to the case of multiple classes and builds integrated toolbox for visualizing the uncertainty. Although the attention has been extensively adopted for the MIL intrepretability, wrong attention is still being generated for the negative bags (see Fig. 2 and Fig. 6). This is not tolerable in the application of medical imaging, where the intrepretability of model is crucial. The reason is, no methods verify the convergence of attention and impose the constraint on correcting attention from the MIL principle. In this work, these issues are carefully resolved and a new probabilistic framework is proposed.

**Orthogonal works** Chen et al. (2022) proposes a large-scale vision transformer solution to simultaneously learn the feature and classifier for WSI. Bayes-MIL freezes the feature extractor during training following a normal setup in MIL. Zhang et al. (2022) proposes a two-stage feature distillation MIL framework for enhancing the performance. Bayes-MIL studies the fundamental interpretability problem in the one-stage MIL framework. See Appendix for a review of uncertainty in DNNs.

## 5 EXPERIMENTS

The proposed methods are evaluated on two standard WSI datasets: CAMELYON16 (Bejnordi et al., 2017) and CAMELYON17 (Bandi et al., 2018). CAMELYON16 contains 400 hematoxylins and eosin (H&E) stained WSI of sentinel lymph node for breast cancer, labeled as normal or tumor classes. CAMELYON17 contains 1000 WSI of the same type, labeled with normal or different stages (pN-stage) of the breast tumor. Since our paper mainly studies the interpretation of MIL, we treat these stages as one class, generating binary slide-level labels (normal and tumors). We leverage the CLAM testbed for the implementation. A ResNet-50 is used for feature extraction, consistent with previous methods. Each result is obtained with *10-fold* splits of training/validation/testing sets, which is a more thorough evaluation than previous papers. Other hyperparameters are listed in the Appendix. The codes are submitted as supplemental. In our evaluation, we consider 3 variants of BayesMIL: 1) Bayes-IL-Vis is the basic Bayesian MIL in Sec. 3.1; 2) Bayes-MIL-SDPR is the model with slide-dependent patch regularizer from Sec. 3.2; 3) Bayes-MIL-Full is the whole model, The hyperparameters are $\lambda_0 = 10^{-8}$ and $\lambda_1 = 10^{-12}$. Results on CAMELYON17 show our method is better than the CLAM baseline in Patch-localization and Slide-classification, shown in the Appendix.

**Patch-level tumor region localization** We first evaluate the patch-level results on tumor region localization. The tumor region localization uses the Patch-Precision, Patch-FROC and Patch-AUC

Table 1: Results on CAMELYON16: (left) Patch-level localization results using Patch-FROC (P-FROC), Patch-Precision (P-Prec.), Patch-AUROC (P-AUC); (right) Slide-level classification results using Slide-Accuracy (S-Acc.), Slide-AUROC (S-AUC) and Slide-Calibration (S-ECE).

| | Patch-level | | | Slide-level | | |
|---|---|---|---|---|---|---|
| | P-Prec. (↑) | P-FROC (↑) | P-AUC (↑) | S-Acc. (↑) | S-AUC (↑) | S-ECE (↓) |
| DSMIL | 0.1030 | 0.4443 | 0.7719 | 0.8682±0.05 | 0.8944±0.06 | 0.3798±0.06 |
| CLAM | 0.6068 | 0.4792 | 0.8839 | 0.8650±0.06 | 0.9177±0.04 | 0.1738±0.06 |
| CLAM-T | 0.7068 | 0.4830 | 0.8884 | — | — | — |
| TransMIL | 0.1726 | 0.4797 | 0.8644 | 0.8837±0.02 | 0.9307±0.04 | 0.1436±0.04 |
| Bayes-MIL-Vis | 0.7140 | 0.4797 | 0.8995 | 0.8825±0.05 | 0.9164±0.06 | 0.1702±0.05 |
| Bayes-MIL-SDPR | 0.7458 | 0.4856 | 0.9001 | 0.8875±0.05 | 0.9432±0.05 | 0.1621±0.05 |
| Bayes-MIL-APCRF | **0.8107** | **0.4919** | **0.9129** | **0.9000**±0.04 | **0.9479**±0.05 | **0.122**±0.04 |

Table 2: The patch-level localization results when using probabilistic attention values, model uncertainty and data uncertainty for our methods, evaluated on CAMELYON16.

| | P-Prec. (↑) | | | P-FROC (↑) | | |
|---|---|---|---|---|---|---|
| | Attn | Model Unc. | Data Unc. | Attn | Model Unc. | Data Unc. |
| Bayes-MIL-Vis | 0.7107 | 0.6999 | **0.7140** | 0.4796 | 0.4788 | **0.4797** |
| Bayes-MIL-SDPR | 0.7445 | 0.7396 | **0.7458** | 0.4781 | **0.4856** | 0.4849 |
| Bayes-MIL-APCRF | 0.8078 | 0.8033 | **0.8107** | 0.4813 | **0.4919** | 0.4879 |

metrics. The Patch-Precision is calculated by averaging the precision of classifying the patches. Each method provides a measure (attention $a_k$ or normalized uncertainty value $\mathcal{U}_k$) for its ROI. For calculating the precision, we compare the measure with three thresholds (0.1, 0.5, 0.9) for all methods and report the best results. Concretely, if $a_k$ or $\mathcal{U}_k$ is greater than the threshold, the patch is predicted as positive, otherwise negative. The Patch-FROC is defined as the average sensitivity (recall) at 6 predefined false positive rates: 1/4, 1/2, 1, 2, 4 and 8 FPs per WSI (Li et al., 2021). The Patch-AUROC evaluates the averaged area under ROC over patches. The ground truth is the doctor's marking the region of the tumor.

For TransMIL, we take the diagonal of attention map from the last multi-head attention module as the measure. We add another baseline, CLAM-T, on CLAM training with temperature $T = [0.2, 0.5, 2, 5]$ on softmax function, $\boldsymbol{a} = \text{softmax}(\cdot, T)$, as a method for manually adjusting the density of localization, and select the best results for different metrics. For the Bayes-MIL-Vis, Bayes-MIL-SDPR and Bayes-MIL-APCRF methods, we take the best results from the normalized data uncertainty, the normalized model uncertainty and the probabilistic attention, with MC integration over 16 samples.

The results for patch-level localization of tumor regions are shown in Tab. 1 (left). The Bayesian modelling of MIL (Bayes-MIL-Vis) generally improves the patch-level visualization over other methods. Tuning the temperature for MIL attention (CLAM-T) could marginally improve localization results, however, it requires exhaustively tuning the temperature. For Bayes-MIL, the precision is improved by a large margin (by 0.1072-0.611), showing the advantage of using uncertainty for localizing positive area. Including SDPR and the approximate CRF improve the patch-level results consistently, and the full model Bayes-MIL-APCRF achieves the best performance on the 3 metrics.

Tab. 2 shows the detailed results with probabilistic attention, model uncertainty and data uncertainty for the three proposed methods. The probabilistic attention benefits from the Bayes-MIL model, making it competitive at tumor region localization. The best results for precision are from the disentangled data uncertainty, indicating the data uncertainty is able to precisely localize the tumor region. The reason is that data uncertainty captures the rareness of features of positive patches indicating anomalies, such as cancerous cells, in the whole dataset. The model uncertainty is better on the P-FROC metric, which evaluates the recall at different false positive rates. This shows that the uncertainty induced from parameter distributions in $\boldsymbol{\pi}$ tends to cover the positive patches better.

**Slide-level evaluation**    We next evaluate the slide-level classification performance. Tab. 1 (right) shows the slide-level performance on CAMELYON16, measured by Accuracy, AUC, and ECE. The results of DSMIL and TransMIL are from their papers, as their reported performance is better than our reimplementation. Compared with DSMIL and CLAM, Bayes-MIL-Vis has a higher accuracy, a lower calibration error and a similar AUC. When including our SDPR, Bayes-MIL-SDPR has better accuracy and AUC than TransMIL.

**Why is Bayes-MIL good at slide classification and ROI localization?**    To understand the reason behind the good performance of the proposed framework, we visualize attention distribution in Fig. 6. As shown in Fig. 6a and Fig. 6b, DSMIL and CLAM generate similar attention distributions for positive and negative slides. For the negative slides, DSMIL still generates a nearly normal

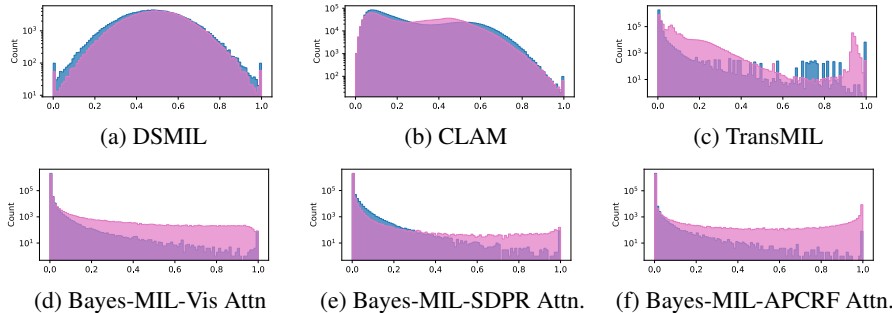

| (a) DSMIL | (b) CLAM | (c) TransMIL |
| (d) Bayes-MIL-Vis Attn | (e) Bayes-MIL-SDPR Attn. | (f) Bayes-MIL-APCRF Attn. |

Figure 6: The log-scale histogram of attention for different methods. The pink and blue curves are for positive and negative slides, respectively. All testing images in CAMELYON16 are used, which contain 2M positive and 2M negative patches.

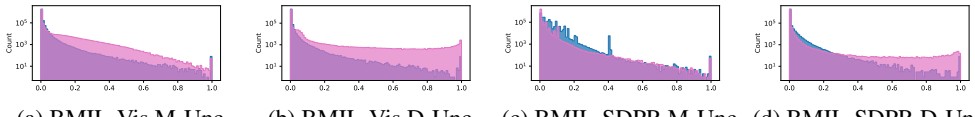

| (a) BMIL-Vis M-Unc. | (b) BMIL-Vis D-Unc. | (c) BMIL-SDPR M-Unc. | (d) BMIL-SDPR D-Unc. |

Figure 7: The log-scale histogram of normalized uncertainty for Bayes-MIL. M and D stand for model and data. APCRF has similar histograms as SDPR, thus not shown for saving space.

distribution of attention, which deviates from the MIL principle. For positive slides, the densities on the positive side ($a \approx 1$) of DSMIL and CLAM are low. Note that the vanilla version BMIL-Vis has different attention distributions for positive and negative slides, but does not generate high density on the positive side for positive slides. Bayes-MIL with SDPR is able to push the attention of negative slides to the negative side ($a \approx 0$, the blue curve in Fig. 6e), while generating a U-shape distribution for the positive slides (red curve in Fig. 6e), which corresponds to our design of SDPR with the ideal case shown in Fig. 4. This design benefits the following aspects:

- For visualization, Bayes-MIL generates concentrated high attention values for positive slides, while only generates correct values ($a \approx 0$) for negative slides.
- For classification, learning correct attention guided by SDPR for the negative slides will benefit the classification of patches in the positive slides. TransMIL captures this attention distribution to some degree, so it performs well in slide-level classification.

Bayes-APCRF performs a smoothing operation on the mean of the stochastic attention nodes. This pushes the histogram to have a more distinct U-shape (the red U-shape curve in Fig. 6f) by aggregating over neighboring positive patches, which benefits the localization and obtains the best visualization scores in Tab. 1.

**Correctly Visualized ROI**  Fig. 2 shows the visualization results on the negative (normal) and positive (tumor) WSI. The related methods all generate high attention for the negative slides, which is against of MIL principle. TransMIL only generates a small area of high attention for negative slides, however the performance on positive slides is poor. *Bayes-MIL is the only method that generate correct values ($a \approx 0$) for the negative slides, while performing the best in localization on positive slides.*

**Measure of uncertainty**  The measure of data uncertainty (Fig. 8b and Fig. 7d) naturally captures the information from the patch distribution. It shows a similar distribution with the attention of Bayes-MIL-SDPR and the ideal case of SDPR in Fig. 4. This might be the reason why data uncertainty has the best performance in localizing the ROI. Furthermore, based on this observation on data uncertainty, the soundness of SDPR is empirically validated.

## 6 CONCLUSION

This paper analyzes the interpretability problem in existing attention-based multiple instance learning (MIL) models. Directly taking attention as a measure of the important instances empirically violates the MIL principle. To address this problem, we propose a probabilistic solution Bayes-MIL that provides new measures for interpretability. To ensure the validity of interpretation in MIL, a regularizer on attention is required. A slide-dependent patch regularizer is proposed for imposing explicit constraints to let the model learn under the MIL principle, which also improves the slide-level performance. The spatial information is further encoded, which improves both slide-level and patch-level performance. The analysis and visualization of data uncertainty distribution further validate the main idea and soundness of SDPR.

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
