# OpenReview forum: "Bayes-MIL: A New Probabilistic Perspective on Attention-based Multiple Instance Learning for Whole Slide Images"
_ICLR.cc/2023/Conference — ICLR 2023 poster_

### Official Review · Reviewer_nxBn · 2022-10-24

**Confidence:** 3
**Correctness:** 4
**Technical Novelty And Significance:** 3
**Empirical Novelty And Significance:** 3
**Recommendation:** 8

**Clarity, Quality, Novelty And Reproducibility:**

* The paper is very well written, well organized and clear.

* The paper contains several novel technical contributions within its bayesian attention-MIL framework for WSI.

* The author provide the codes, which should help reproduce the approach


**Strength And Weaknesses:**

Pros:

* The method improves SOTA both on interpretability (the high attention patches correspond to tumor areas) and accuracy.

* The spatial correlation of patches is learned by a CRF, which conforms to the tendency of tumors to locally grow.

* The contribution of each module is clearly reported and show that each of them provides additional improvements.

Cons:

* The feature extractor is simply a resnet50 (imagenet). For the WSI task, other works have shown that self-supervised models trained on WSI data do have superior results.

* There are no speed information given for training and inference. How long does it take to classify a slide on average ? How long does it take to train the feature extractor, the MIL model ? Also a pseudo-code of the algorithm would help.


**Summary Of The Paper:**

This paper proposes a method for attention-bases multi-instance-learning (MIL) on digital pathology whole slides (WSI) for tumor classification. The method improves upon SOTA MIL methods on WSI by approaching the problem using a probabilistic framework. A drawback of current attention-based WSI MIL methods is that attention is not focused on positive (tumor) instances only. The proposed approach ensures that attention is focused on positives instances, making the results more interpretable as high attention areas match tumor areas. A Bayesian modeling of MIL provides calibrated uncertainty at the patch level. A variational inference framework regularizer produces dense localization of positive patches. Finally the spatial correlations are captures by a conditional random field CRF. The empirical section compares the proposed approach to SOTA MIL WSI approaches such as DSMIL, TransMIL on two public datasets, showing significant improvements. The results are also ablated showing the contribution of individual modules.


**Summary Of The Review:**

A well-written paper that proposes a novel way to improve on interpretability of WSI classification, while also improving SOTA accuracy on 2 large public datasets.

---

> ### Author Response · Authors · 2022-11-15
> **Discussion of feature extractors and efficiency problems.**
>
> >The feature extractor is simply a resnet50 (imagenet). For the WSI task, other works have shown that self-supervised models trained on WSI data do have superior results.
>
> Thanks for the practical suggestion for improving the general performance. Note that we focus more on the understanding of interpretability problem in MIL and try to solve it from a probabilistic perspective. The feature extractor is assumed to be fixed (a pretrained ResNet-50). Using self-supervised models is an orthogonal direction for improving the performance of MIL to Bayes-MIL. Nonetheless, using a stronger feature extractor makes patch-level features more linear-separable. We have been exploring different feature extractors like masked autoencoder (MAE), SimCLR and MoCov3. We will update the results before rebuttal deadline if time permits.
>
> >There are no speed information given for training and inference. How long does it take to classify a slide on average ? How long does it take to train the feature extractor, the MIL model ? Also a pseudo-code of the algorithm would help.
>
> We show the training time per epoch and testing time per slide in the following table.
>
> |                              | CLAM  | TransMIL | DSMIL | Bayes-MIL |
> |------------------------------|-------|----------|-------|-----------|
> | Training Time (minute per epoch) | 0.606 | 1.8      | 0.824 | 1.217     |
> | Testing Time (second per slide) | 0.062 | 5.857      | 0.045 | 0.185     |
>
> Here, Bayes-MIL uses an efficiency-optimized approximate CRF module instead of directly performing convolution. The library will be released.
> The training time of feature extractor is not considered as a pre-trained model is adopted, like CLAM, TransMIL and etc.
>
> The pseudo codes are updated in Section C of the appendix.

---

### Official Review · Reviewer_ndc8 · 2022-10-30

**Confidence:** 4
**Correctness:** 3
**Technical Novelty And Significance:** 2
**Empirical Novelty And Significance:** 3
**Recommendation:** 6

**Clarity, Quality, Novelty And Reproducibility:**

This paper studies an interesting problem in MIL-based WSI analysis. While some details about the method design are not clear and the experimental part is a little weak.

**Strength And Weaknesses:**

Strength:
1. The discussed problem is interesting and important for AI-assisted pathology diagnosis.
2. The proposed slide-dependent patch regularizer (SDPR) explicitly encodes the basic assumption of MIL into the training process, which is a long-ignored issue in weakly-supervised learning for WSI analysis.
3. Good performance on two breast cancer datasets in normal/tumor classification tasks.

Weakness:
1. The authors should discuss more about the relationship between the WSI pathology diagnosis and MIL. It is easy to understand that the normal/tumor classification in WSI diagnosis followed the basic assumption of MIL and can benefit from the proposed SDPR. However, other tasks in WSI diagnosis may not meet the same requirements. For example, both tumor subtype classification with more than two classes and tumor stage classification tasks cannot be dealt with the proposed Bayes-MIL and SDPR. Does this mean that the current solution is limited in the simplest positive/negative setting in MIL?
2. As for the tumor/normal classification, it is far more important to localize the ROI in tumor WSI instead of the normal WSI. However, in the Tumor WSIs in both Fig 2. And Fig 8., CLAM seems to provide a more accurate (the highlighted regions have better shape and is denser) visualization than the proposed method.
3. Is it ok to conclude that in previous works, high attention weights are used to indicate it associated instances are cancerous image patches? If the slide-level prediction for a WSI is normal (without cancer), how do we interpret the high attention weights?
4. From Table 1., we can see that without SDPR and APCRF, the proposed method is not better than previous attention-based works. Does this mean the data uncertainty is not necessary? Can we apply the SDPR and APCRF to previous attention-based works and achieve satisfactory improvement? More ablation study is needed to validate the motivation and effectiveness of the proposed Bayes-MIL.
5. The experiments are limited to the breast cancer datasets and normal/tumor classification task.
6. The std of the experiments is not provided in the paper.


**Summary Of The Paper:**

The paper studies multiple instance learning for WSI analysis. They pointed out that the interpretability of previous attention-based MIL methods for WSI diagnosis was either untrustworthy or unsatisfactory and proposed Bayes-MIL to induce patch-level uncertainty as a new measure of MIL interpretability. To impose constraints derived from the MIL assumption, they further designed a slide-dependent patch regularizer. Last, they applied an approximate convolutional conditional random field to encode the spatial information in WSI. The proposed method was evaluated on public datasets.

**Summary Of The Review:**

This paper discusses an interesting problem in MIL-based WSI analysis, while the current version need further improved in regard of clarification and evaluation.

After response:

The new clarification and experiments have partially addressed my concerns. Therefore,  I would raise my score to 6: marginally above the acceptance threshold.

---

> ### Author Response · Authors · 2022-11-15
> **Discussion of other tasks for histopathological diagnosis.**
>
> (Response to Weakness 1)
>
> Appreciate the practical suggestions from the pathological perspective.
>
> The proposed Bayes-MIL studies the fundamental interpretability problems of attention-based MIL [Ilse et al., 2018], which is by default a binary classification setting at the slide level. For validation, we have tried the largest dataset we could found under this setting, Camelyon16 and Camelyon17 in two merged classes. Tumor subtype classification and tumor stage classification are important applications for AI-assisted histopathology diagnosis. *They are not the exact attention-based MIL problems and not helpful for our empirical analysis*, which will be explained below. *However, our method is certainly useful for these cases.* We present the details on how to extend our methods to these problems and give some preliminary results.
>
> I. For tumor subtype classification, the reason that we did not validate our method on this data type is, it is not an exact attention-based MIL problem as normally negative slides are not provided (e.g., TCGA-RCC and TCGA-NSCLC), which violates the general assumption of MIL. This application could not be understood from MIL perspective and the theoretical analysis could not be applied.
>
> Empirically, the proposed Bayes-MIL is suitable for tumor subtype classification which might not include negative classes, with simple modifications. Taking TCGA-NSCLC as an example, this dataset includes two classes. In this case, the MIL model looks for the classification boundary between positive slides. Although this task cannot be understood from MIL perspective, the method of Bayes-MIL still applies. For Bayes-MIL-Vis, the uncertainty could be extracted with the same pipeline. The prior of Bayesian neural network is a special case of SDPR where every slide is given the same regularizer $\mathcal{LN}(\mu_1, \sigma_1)$. The CRF part remains the same.
>
> II. For tumor stage classification with multiple stages, it is an MIL problem with multiple classes, as negative slides are provided (e.g., the original Camelyon17). Empirically, the proposed framework could be adapted in following way:
>
> At the patch level, the attention still distinguishes the positive and negative patches. Tumors at different stages could be regarded as the positive slides, given the positive SDPR regularizer $\mathcal{LN}(\mu_1, \sigma_1)$. The negative slides are given the negative SDPR regularizer $\mathcal{LN}(\mu_0, \sigma_0)$. The hyper-parameters are the same as presented in the manuscript.
>
> Note that, $\mathcal{LN}(\mu_1, \sigma_1)$ could be further customized by domain experts (e.g., doctors), by giving different stages different hyper-parameters $\{(\mu_{11}, \sigma_{11}), (\mu_{12}, \sigma_{12}), (\mu_{13}, \sigma_{13}), ...\}$. In this way, the expert knowledge could be quantified and encoded in the MIL model training.
>
> At the slide level, the Bayes-MIL model could use a multi-class classification head (e.g., softmax) to perform the multi-class classification.
>
> We provides some preliminary testing results on TCGA-NSCLC and the comparisons with other methods. The comparison results on Camelyon17 tumor stage classification are provided as well. We will update the other experiments before rebuttal deadline if time permits.
>
> Results on TCGA-NSCLC:
> | Methods         | AUC         | ACC         | ECE         |
> |-----------------|-------------|-------------|-------------|
> | CLAM            | 0.941956617 | 0.863983536 | 0.169745733 |
> | Scaling ViT     | 0.951580805 | 0.882097156 | N/A         |
> | TransMIL        | **0.9603**      | 0.8835      | N/A         |
> | Bayes-MIL-Enc   | 0.944047204 | 0.889345855 | 0.164659107 |
> | Bayes-MIL-APCRF | 0.945077016 | **0.896553273** | **0.157494113** |
>
> Results on Camelyon17 tumor stage classification:
> | Methods         | AUC         | ACC         | ECE         |
> |-----------------|-------------|-------------|-------------|
> | CLAM            | 0.7803 | 0.6 | 0.4138 |
> | Bayes-MIL     | **0.8070** | **0.64** | **0.4017**        |
>
> [Ilse et al., 2018] Maximilian Ilse, Jakub Tomczak, and Max Welling. Attention-based deep multiple instance learning. In International conference on machine learning, pp. 2127–2136. PMLR, 2018.

---

> > ### Comment · Reviewer_ndc8 · 2022-11-29
> > **Thanks for your respone**
> >
> > Thanks for your effort in addressing my concerns. Further clarification and new experiments have addressed my previous concerns, and I will raise my score to 6: marginally above the acceptance threshold.

---

> ### Author Response · Authors · 2022-11-15
> **Importance of avoiding false positive cases and revised visualization.**
>
> (Response to Weakness 2)
>
> We disagree with the claim. For the negative slides, providing the correct interpretation is important as well. First, this makes sure that MIL works correctly under its original theoretical assumption. Second, for the histopathology diagnosis application, much False Positive (FP) interpretations would bias the judgement of doctor. Performing a surgery on a healthy person is *NOT* acceptable. The visualization presented to doctor as an evidence of AI diagnosis should contain correct interpretations for both positive and negative slides, in order to be trustworthy. From the visualization in Fig. 2, we can see that the wrong interpretation issue is severe on CLAM compared with other methods.
>
> We show more numerical results of the patch-level accuracy in *normal slides only (Camelyon16)*.
>
> | Methods         | ACC (threshold=0.1) | ACC (threshold=0.3) | ACC (threshold=0.5) |
> |-----------------|---------------------|---------------------|---------------------|
> | DSMIL           | 0.0058              | 0.1188              | 0.5547              |
> | CLAM            | 0.2758              | 0.6650              | 0.8395              |
> | TransMIL        | 0.9846              | 0.9898              | 0.9920              |
> | Bayes-MIL-APCRF | **0.9986**              | **0.9996**              | **0.9998**              |
>
> Under different thresholds, Bayes-MIL-APCRF consistently presents high accuracy while other methods (e.g., CLAM) do not provides satisfactory results.
>
> --------
>
> The previous visualizations might be misleading as it uses different colors. We updated the visualizations in Fig. 2 with only one color. CLAM provides high attention values outside the positive area for positive slides, which is not satisfactory.
>
> For the positive slides, outside the positive region, CLAM also provides high attention values (~0.4) for the negative regions, which by the MIL assumption should have low attention values. Empirically, as the optimal threshold (for determining positive and negative slides) is unknown, it is not desirable to have vague values like 0.4. The updated visualization in Fig. 2 shows the interpretation of CLAM is un-satisfactory, while Bayes-MIL do not suffer from this. Bayes-MIL provides accurate localization reflected by both experimental results (precision and FROC) and visualization. Fig. 9 shows the segmented results with a certain threshold, which further shows the CLAM visualization might cause a severely wrong interpretation.
>
> We visualize the results of the patch-level interpretation in *tumor slides only (Camelyon16)* in Fig.8, appendix. It could be concluded that Bayes-MIL presents consistently decent performance under different threshold while CLAM should explore its optimal threshold.

---

> ### Author Response · Authors · 2022-11-15
> **Responses to other comments.**
>
> (Responses to weakness 3, 4, 5, 6)
>
> ### Understanding of high attention weights in normal slides. (W3)
>
> >Is it ok to conclude that in previous works, high attention weights are used to indicate it associated instances are cancerous image patches? If the slide-level prediction for a WSI is normal (without cancer), how do we interpret the high attention weights?
>
> First of all, *it is not OK to conclude in such way*, which we did not. In the analysis, we show in the case of binary classification (the same assumption in "Attention-based deep multiple instance learning"), under mild assumption, high attention weights will converge to indicate the associated instances are cancerous. However, the convergence of attention depends on initialization and previous works did not address that. Thus, it is not OK to conclude in such way. To determine perfect initialization is not easy, we choose to use uncertainty as another indicator for localization. We further propose the SDPR, leveraging the variational inference framework, to enforce the correct localization.
>
> Previous works use deterministic neural network, thus there is no source of uncertainty in the model. The high attention weights in normal slides are mis-classifications for patches. In Bayes-MIL, the model uncertainty is introduced by turning weights to be probabilistic nodes. Then uncertainty over attention is then induced from model uncertainty, by Monte-Carlo integration in validation/testing time. The high attention weights in normal slides indicate which patch the model is uncertain about or which patch is rarely seen in the training data.
>
> ### Understanding of the experimental results of instance-level disentangled uncertainty - Bayes-MIL-Vis (W4)
> >From Table 1., we can see that without SDPR and APCRF, the proposed method is not better than previous attention-based works. Does this mean the data uncertainty is not necessary? Can we apply the SDPR and APCRF to previous attention-based works and achieve satisfactory improvement? More ablation study is needed to validate the motivation and effectiveness of the proposed Bayes-MIL.
>
> We disagree with the reviewer's exposition of our experimental results. The data uncertainty and model uncertainty are proposed for a new metric for the patch-level localization. If we look at the patch-level results, Bayes-MIL-Vis outperforms all previous methods on P-Prec and P-AUC, and most previous methods on P-FROC. The only exception is that CLAM-T outperforms Bayes-MIL-Vis which is our enhancement on CLAM using temperature scaling. This method requires training another temperature parameter on the validation set, which requires extra-training time.
>
> On slide-level results, only TransMIL has better performance than Bayes-MIL-Vis. However, it is worth to note that our uncertainty decomposition in Bayes-MIL-Vis does not necessarily improves the slide-level performance. There is no expectation on that. But SDPR and APCRF would do.
>
> The SDPR is a prior of Bayesian neural network under variational inference framework. APCRF is designed for encoding the spatial information by a probabilistic conditional attention. These two methods require the MIL model to be probabilistic, which is exactly what we proposed in this paper. The previous methods are deterministic, thus our methods are not directly applicable to them.
>
> ### More experiments and std (W5, W6)
> We add the experimental results on tumor subtype classification on TCGA-NSCLC and tumor stage classification on CAMELYON17. On tumor subtype classification, evaluations show Bayes-MIL outperforms CLAM in all metrics. Bayes-MIL outperforms the ViT-based baseline in accuracy and calibration. On tumor stage classification, Bayes-MIL outperforms CLAM in all metrics.
>
> The std for slide-level prediction on Camelyon16, Camelyon17 and TCGA-NSCLC are updated in the paper.

---

### Official Review · Reviewer_c5qM · 2022-10-31

**Confidence:** 3
**Correctness:** 4
**Technical Novelty And Significance:** 3
**Empirical Novelty And Significance:** 1
**Recommendation:** 8

**Clarity, Quality, Novelty And Reproducibility:**

The paper is well-written and well-organized. For reproducibility, the code has been provided. The novelty of the paper is in the combination of different elements that improve the SOTA.


**Strength And Weaknesses:**

Strength
The proposed approach improves upon existing MIL approaches on different aspects: It improves the performance at slide and patch level. It improves the interpretability by introducing calibrated model uncertainty.
The paper uses a 10-fold cross validation in the evaluation, which is not frequently seen in the literature due to the computational cost.

Weakness
The scalability of the approach is not discussed. How much Bayes-MIL is slower than baselines in training and inference time.


**Summary Of The Paper:**

The paper proposes interpretable models in multiple instance learning for WSI classification. The authors identify problems in using directly attention mechanism for interpretation. The experiments conducted indicates that high attention values are still generated in negative slides, indicating positive patches. To address this issue, the authors propose Bayesian MIL incorporating 1) probabilistic instance-wise attention module for uncertainty visualization 2) slide-dependent patch regularizer for learning the correct attention distribution 3) an approximate convolutional conditional random field for encoding spatial information. The experiments on two WSI datasets showed that the proposed Bayes-MIL outperforms baselines in path and slide levels metrics.

**Summary Of The Review:**

The paper introduces a new approach for MIL problem which brings uncertainty measure in the model and improves interpretability. This is highly relevant for pathology applications where annotations are available only at slide level.

---

> ### Author Response · Authors · 2022-11-15
> **Discussion of efficiency.**
>
> >The scalability of the approach is not discussed. How much Bayes-MIL is slower than baselines in training and inference time.
>
> We appreciate the comments and feedback. For the efficiency issue, we show the training time per epoch in the following table.
>
> |                              | CLAM  | TransMIL | DSMIL | Bayes-MIL |
> |------------------------------|-------|----------|-------|-----------|
> | Training Time (minute per epoch) | 0.606 | 1.8      | 0.824 | 1.217     |
> | Testing Time (second per slide) | 0.062 | 5.857      | 0.045 | 0.185     |
>
> Bayes-MIL provides acceptable training and testing time compared with other methods, being more efficient than TransMIL. Here, Bayes-MIL uses an efficiency-optimized approximate CRF module instead of directly performing convolution.

---

### Author Response · Authors · 2022-11-15
**General responses**

We thank all reviewers for the constructive comments.

We conclude the mentioned major problems and the corresponding answers/clarifications/explanations in the following parts:

1. Efficiency (Reviewer c5qM and Reviewer nxBn): The missed running time information are presented in the responses and updated in the manuscript (Table 3, Appendix). The pseudo codes (Reviewer nxBn) are updated as well (Section C, Appendix). By using an efficient variational inference design and implementation, we keep the running time of Bayes-MIL between CLAM and TransMIL.

2. Other tasks like tumor subtype classification and tumor stage classification (Reviewer ndc8): The proposed Bayes-MIL studies the fundamental interpretability problems of attention-based MIL, which is by default a binary classification setting at the slide level. Tumor subtype classification is not an exact MIL problem, while tumor stage classification is a multi-class extension of the MIL problem. They are not helpful for our empirical analysis for understanding the functionality of probabilistic MIL. We still show our method could be useful for the tumor subtype classification and tumor stage classification tasks and outperforms the baselines (Table 6 and Table 7 Appendix).

3. Importance of avoiding false positive cases / comparing visualizations with CLAM (Reviewer ndc8): We argue the correct interpretation on normal slides is important as well, for making the MIL work correctly under its original assumption and avoiding biasing the judgement of users (e.g., doctors).
Previous visualizations might be misleading as it uses different colors in one figure. We fixed Fig. 2 and show Bayes-MIL can locate the tumor region more precisely. Numerical results for both normal slide only and tumor slides only are presented for more solid illustrations (Table 4, Figure 8, Appendix).

4. Extending SDPR and APCRF to existing MIL methods (Reviewer ndc8): SDPR and APCRF could work when the MIL method is probabilistic. As the existing methods are deterministic, it’s not directly applicable.

The changes are highlighted in red text in the revised manuscript.

We are happy to swiftly respond and clarify any remaining question.

Sincerely,
Authors

---

### Author Response · Authors · 2022-11-24
**Look forward to post-rebuttal feedbacks!**

Dear AC and reviewers:

Thanks again for all of your constructive suggestions, which helped us improve the quality and clarity of the paper!

Our rebuttal has been posted for a while. We have not heard any post-rebuttal response yet. Please do not hesitate to let us know if there are any additional clarifications that we can provide, as we would like to convince you of the merits of the paper.

We appreciate your responses. Thanks!

Sincerely,
Authors

---

> ### Comment · Reviewer_nxBn · 2022-12-07
> **Thank you for your response**
>
> I appreciate the authors' thoughtful and extensive response to my comments as well as those from the other reviewers. I keep my good score.

---

### Decision · Program_Chairs · 2023-01-20

**Decision:**

Accept: poster

**Justification For Why Not Higher Score:**

While the reviewers agreed that the paper has sufficient novelty, they also comment that the paper is not an outstanding one that might attract many researchers. The applied area is relatively narrow.

**Justification For Why Not Lower Score:**

The rebuttal has addressed the reviewers' concerns and there is a consensus that the paper has sufficient novelty.

**Metareview: Summary, Strengths And Weaknesses:**

The paper presents a new approach for multiple instance learning by integrating uncertainty into MIL.
The strength:
1) It has two main novelties to encode the underlying logic of MIL into the training process and encode spatial information via approximation to convolutional CRF.
2) The results justify the effectiveness of the proposed methods.
Weakness: The validation is done in digital pathology and its usefulness in other areas is not clear.




**Note From Pc:**

if the above contains the word "oral" or "spotlight" please see: "oral" presentation means -> notable-top-5% and "spotlight" means -> notable-top-25%. As stated in our emails, we are disassociating presentation type from AC recommendations

**Summary Of Ac-Reviewer Meeting:**

The rebuttal has addressed most of concerns from the reviewers.
It may not be an outstanding one for oral, but it is sufficient for poster.